# Influence of thromboembolic events in the prognosis of COVID-19 hospitalized patients. Results from a cross sectional study

**Francisco Purroy**[1,2,3,4☯‡]*, **Gloria Arqué**[1,5☯‡]

**1** Clinical Neurosciences Group, Institut de Recerca Biomèdica de Lleida (IRBLleida), Lleida, Spain, **2** Stroke Unit, Department of Neurology, Hospital Universitari Arnau de Vilanova, Lleida, Spain, **3** Medicine Department, Universitat de Lleida, Lleida, Spain, **4** Medicine Department, Institut de Recerca Biomèdica de Lleida (IRBLleida), Universitat de Lleida, Lleida, Spain, **5** Experimental Medicine Department, Institut de Recerca Biomèdica de Lleida (IRBLleida), Universitat de Lleida, Lleida, Spain

☯ These authors contributed equally to this work.
‡ These authors are joint senior authors on this work.
* fpurroygarcia@gmail.com

**Data Availability Statement:** All relevant data are within the paper and its Supporting Information files.

**Funding:** The authors received no specific funding for this work.

## Abstract

### Background

COVID-19 may predispose to both venous and arterial thromboembolism event (TEE). Reports on the prevalence and prognosis of thrombotic complications are still emerging.

### Objective

To describe the rate of TEE complications and its influence in the prognosis of hospitalized patients with COVID-19 after a cross-sectional study.

### Methods

We evaluated the prevalence of TEE and its relationship with in-hospital death among hospitalized patients with COVID-19 who were admitted between 1st March to 20th April 2020 in a multicentric network of sixteen Hospitals in Spain. TEE was defined by the occurrence of venous thromboembolism (VTE), acute ischemic stroke (AIS), systemic arterial embolism or myocardial infarction (MI).

### Results

We studied 1737 patients with proven COVID-19 infection of whom 276 died (15.9%). TEE were presented in 64 (3.7%) patients: 49 (76.6%) patients had a VTE, 8 (12.5%) patients had MI, 6 (9.4%%) patients had AIS, and one (1.5%) patient a thrombosis of portal vein. TEE patients exhibited a diffuse profile: older, high levels of D-dimer protein and a tendency of lower levels of prothrombin. The multivariate regression models, confirmed the association between in-hospital death and age (odds ratio [OR] 1.12 [95% CI 1.10–1.14], p<0.001), diabetes (OR 1.49 [95% CI 1.04–2.13], p = 0.029), chronic obstructive pulmonary disease (OR 1.61 [95% CI 1.03–2.53], p = 0.039), ICU care (OR 9.39 [95% CI 5.69–15.51], p<0.001), and TTE (OR 2.24 [95% CI 1.17–4.29], p = 0.015).

**Competing interests:** The authors have declared that no competing interests exist.

**Abbreviations:** AIS, Acute Ischemic Stroke; CI, Confidence Interval; COPD, Chronic Obstructive Pulmonary Disease; COVID-19, Coronavirus Disease 2019; ICU, Intensive Care Unit; OD, Odds Ratio; TEE, Thromboembolism Event; VTE, Venous Thromboembolism.

## Conclusions

Special attention is needed among hospitalized COVID-19 patients with TTE and other comorbidities as they have an increased risk of in-hospital death.

## Introduction

The coronavirus disease of 2019 (COVID-19) is a viral illness caused by the severe acute respiratory syndrome coronavirus 2 (SARS-CoV2) that is now considered a pandemic by the World Health Organization [1–3]. Although, initial efforts have been focused on the diagnosis and treatment of severe pneumonia with vital compromise [1, 3], a spectrum of extra respiratory symptoms and signs generated by the infection itself have been verified. Among those symptoms, we can stand out that COVID-19 infection might predispose patients to thrombotic disease, both in the venous and arterial circulations mediated by inflammation, endothelial dysfunction, thrombin generation, platelet activation and stasis [4, 5]. Recently, it was reported a 31% prevalence of thrombotic complications in ICU patients with COVID-19 infections in Dutch intensive care units [6]. Coagulopathy and over disseminated intravascular coagulation appear to be associated with high mortality rates. Among the coagulation parameters, D-dimer elevation was the strongest independent predictor of mortality [7], and high levels of D-dimer have been observed in patients admitted to intensive care units [1]. Non-survivors have shown significantly higher levels of plasma D-dimers and fibrin degradation products, increased prothrombin times and activated partial thromboplastin times compared to survivors [8]. Moreover, the metallopeptidase enzyme of angiotensinogen converter 2 (ACE2), identified as the cellular receptor for the coronavirus, is expressed in alveolar epithelial cells and in endothelial cells [9]. The prothrombotic predisposition seems to be developed more intensely from the tenth day of infection [10], and it would also be related to the inflammatory effect of the COVID-19 infection.

The relationship between inflammation and ischemic episodes is already described [11, 12] and COVID-19 associated ischemic strokes are more severe with worse functional outcome and higher mortality than non-COVID-19 ischemic strokes patients [13, 14]. COVID-19 has a number of important cardiovascular implications [15–17]. There is a high prevalence of cardiovascular disease among patients with COVID-19 and acute cardiac injury is commonly observed in severe cases. Patients with prior risk factors are at higher risk for adverse events from COVID- 19 and worse prognosis [18, 19].

Here, we evaluated the prevalence of thromboembolism event (TEE) in all COVID-19 patients admitted to the network of 16 Hospitals in Spain (HM Hospitals). The influence of the composite outcome in the risk of death during the admission was also determined.

## Material and methods

### Data source

This is a cohort study based on anonymized clinical dataset provided by HM Hospitales, which includes sixteen hospitals from all over Spain (HM Delfos, HM Sant Jordi, HM nens, HM modelo, HM Belén, HM Rosaleda, HM la esperanza, HM San Francisco, HM regla, HM Madird, HM Montepríncipe, HM Torrelodones, HM Sanchinarro, HM Nuevo Belén, HM Puerta del Sur and HM Vallés), with detailed hospital admission information for COVID-19 patients from March 1st to April 20th 2020. Database was accessed by the research on August

2020. Strengthening the Reporting of Observational Studies in Epidemiology (STROBE) guidelines were applied [20].

## Study participants

A dedicated electronic medical record data extraction protocol was developed to identify all patients with confirmed COVID-19 (defined as a positive SARS-CoV2 reverse-transcriptase polymerase chain reaction test by nasopharyngeal/oropharyngeal swab or sputum specimen). Patients whose age was under 18 years were excluded (n = 11) (Fig 1). We retrieved the primary International Classification of Diseases (ICD10, 10th Revision, Clinical Modification) codes for each patient. For the selected patients, clinical characteristics, demographic variables, admission at ICU, laboratory determinations, coexisting conditions and vascular risk factors based on ICD10 were collected for further analysis. Patients without ICD10 codification were also excluded (n = 131) (Fig 1).

The primary outcome was the occurrence of TEE defined by venous thromboembolism (VTE), acute ischemic stroke (AIS), systemic arterial embolism or myocardial infarction. The relationship of TEE with the endpoint of in-hospital discharge or exitus was also evaluated.

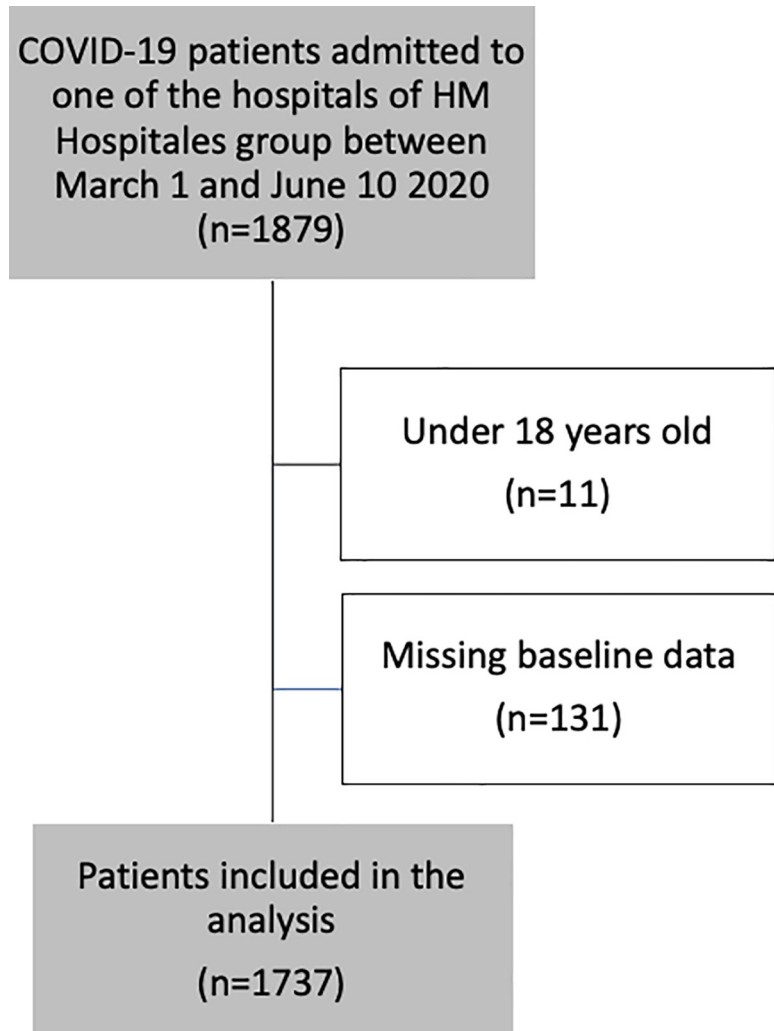

**Fig 1. Patients flowchart.**

## Data analysis

Categorical variables were presented as numbers and percentages and continuous variables as mean and standard deviation if they were normally distributed or median and interquartile range if they were not normally distributed. Variables related to the composite outcome and death were identified. We compared means for continuous variables by using independent group t-tests when the data was normally distributed; otherwise, we used the Mann-Whitney test. Proportions for categorical variables were compared using the chi-square ($\chi2$) test. Fisher's exact test was used in the analysis of contingency when the sample sizes were small. Multivariate logistic regression analysis was used to identify predictors of death, in which variables showing p≤0.10 on a univariate testing were included. Forward selection model with likelihood ratio (LR) was used, first demographic and clinical characteristics (model 1) were added and vital signs (model 2) were added afterwards, to analyze the single best improvement of the model. All the tests were 2-sided and were run at a statistical significance level of p≤0.05. Data was analyzed using SPSS Statistics software, version 20 (IBM SPSS, Chicago, IL, USA).

The study was approved by the Ethics committee of the HM Hospitals (approval number 20.03.1573-GHM).and adhered to the policy for protection of human subjects according to Declaration of Helsinki. As the information was obtained from a de-identified public database written informed consent for our specific study from all study participants was not needed. The data were analyzed anonymously.

## Results

A total of 1737 patients (Fig 1) were included in the current analysis, all of them were COVID-19 positive. Of those, 64 (3.7%) patients had a TEE: 49 (76.6%) patients had a VTE, 8 (12.5%) patients had an acute ischemic heart disease event, 6 (9.4%%) patients had an AIS, and one (1.5%) patient a thrombosis of portal vein. Median time from admission to discharge was 7 (interquartile range [IQR], 4.0–10.0) days. The main diagnosis at admission was pneumonia or respiratory symptoms present in 1606 (92.5%) patients. Hypertension was the main vascular risk factor presented in 647 (37.2%) patients. 117 (6.7%) patients required intensive care. 276 patients died after hospital's admission, corresponding to an in-hospital mortality rate of 15.9%. Median time until death was 5.0 days (IQR 3.0–9.0) days.

### Variables associated with thromboembolic events

Dataset was clustered by the absence/presence of TEE (Table 1). The group of TEE was characterized by older patients (mean age 68.9 [standard deviation, SD:14.0] vs. 65.3 [SD:16.7] years, p = 0.097). TEE patients had signicantly less pneumonia or respiratory symptoms at admission than non-TEE patients. (75.0% versus 93.1%, p<0.001). Mortality risk during admission was significant higher in TEE patients than in non-TEE patients (29.7% [95% CI 18.5–40.9] vs. 15.4% [95% CI 13.7–17.1], p = 0.002). No descriptive variables nor previous coexisting conditions or vascular risk factors were significantly associated with TEE. D-dimer levels were significantly higher among TEE patients than in non-TEE patients (median 6679.0 [IQR 661.0–11297.0] vs. 742.5 [IQR 425.0–1496.0] ng/mL, p = 0.009); and prothrombin time showed a lower tendency in TEE patients (median 12.6 [IQR 11.9–13.2] vs. 13.3 [IQR 12.4–14.6] seconds, p = 0.056) (Table 1).

### Associations of in-hospital mortality

Non-survivor patients were significantly older than survivor patients (mean 80.7 [SD 9.7] vs. 62.6 [SD 9.7] years, p<0.001). A higher prevalence of risk factors was also presence in the

**Table 1. Description and frequency of baseline characteristics, comorbidity, clinical, laboratory variables in patients with thromboembolic events.**

| | | All | Non-TEE | TEE | p-value[a] |
|---|---|---|---|---|---|
| n (%) | | 1737 | 1673 (96.3) | 64 (3.7) | - |
| **Age,** mean (SD), years | | 65.5 (16.7) | 65.3 (16.7) | 68.9 (14.0) | 0.097 |
| **Sex female** | | 686 (39.5) | 660 (39.5) | 26 (40.6) | 0.850 |
| **Hypertension** | | 647 (37.2) | 622 (37.2) | 25 (39.1) | 0.760 |
| **Diabetes** | | 263 (15.1) | 255 (15.2) | 8 (12.5) | 0.548 |
| **Hyperlipidemia** | | 390 (22.5) | 376 (22.5) | 14 (21.9) | 0.910 |
| **Chronic kidney disease** | | 64 (3.7) | 63 (3.8) | 1 (1.6) | 0.358 |
| **Congestive heart failure** | | 38 (2.2) | 35 (2.1) | 3 (4.7) | 0.164 |
| **Atrial fibrillation** | | 72 (4.1) | 70 (4.2) | 2 (3.1) | 0.677 |
| **Previous ischemic heart disease** | | 91 (5.2) | 89 (5.3) | 2 (3.1) | 0.439 |
| **Previous cerebrovascular disease** | | 15 (0.9) | 14 (0.8) | 1 (1.6) | 0.538 |
| **Pneumonia or respiratory symptoms** | | 1606 (92.5) | 1558 (93.1) | 48 (75.0) | **<0.001** |
| **Chronic obstructive pulmonary disease** | | 119 (6.9) | 112 (6.7) | 7 (10.9) | 0.187 |
| **Ex-smoker** | | 258 (14.9) | 246 (14.7) | 12 (18.8) | 0.372 |
| **Current smoker** | | 52 (3.0) | 50 (3.0) | 2 (3.1) | 0.950 |
| **Management** | | | | | |
| **ICU admission** | | 117 (6.7) | 110 (6.6) | 7 (10.9) | 0.172 |
| **Days of hospitalization,** median (IQR) | | 7.0 (4.0–10.0) | 7.0 (4.0–10.0) | 7.5 (4.0–13.0) | 0.118 |
| **Death** | | 276 (15.9) | 257 (15.4) | 19 (29.7) | **0.002** |
| **Vital signs** | | | | | |
| **Systolic blood pressure,** mean (SD) mmHg | N = 1099 | 131.5 (21.5) | 131.4 (21.4) | 132.2 (22.3) | 0.799 |
| **Diastolic blood pressure,** mean (SD) mmHg | N = 1104 | 76.5 (33.0) | 76.4 (33.6) | 78.6 (14.2) | 0.643 |
| **Oxygen saturation,** mean (SD) | N = 1385 | 92.3 (7.0) | 92.3 (7.0) | 91.4 (7.4) | 0.357 |
| **Laboratory determinations** | | | | | |
| **D-dimer,** median (IQR) ng/mL | N = 257 | 822.5 (450.5–1745.5) | 742.5 (425.0–1496.0) | 6679.0 (661.0–11297.0) | **0.009** |
| **Alanine aminotransferase,** median (IQR) U/L | N = 277 | 28.2 (17.0–49.0) | 27.1 (17.0–46.7) | 37.0 (23.8–67.1) | 0.343 |
| **Aspartate aminotransferase,** median (IQR) U/L | N = 284 | 34.0 (23.8–57.1) | 33.0 (23.3–56.3) | 43.5 (25.0–78.2) | 0.265 |
| **Creatinine,** median (IQR) mg/dL | N = 304 | 0.9 (0.7–1.1) | 0.9 (0.7–1.1) | 0.8 (0.7–0.9) | 0.232 |
| **Hemoglobin,** median (IQR) g/dL | N = 297 | 13.6 (12.2–14.8) | 13.7 (12.5–14.8) | 14.1 (11.6–15.9) | 0.696 |
| **Leukocytes,** median (IQR) x10e3/µL | N = 308 | 6.6 (5.0–9.5) | 6.4 (4.9–8.9) | 6.8 (5.0–9.1) | 0.857 |
| **Neutrophil count,** median (IQR) x10e3/µL | N = 308 | 4.8 (3.3–7.6) | 4.6 (3.3–7.1) | 4.9 (3.3–7.6) | 0.772 |
| **Platelet count,** median (IQR) x10$^3$ per uL | N = 302 | 218.5 (168.3–299.5) | 214.0 (165.5–301.0) | 240.0 (209.5–271.5) | 0.493 |
| **C reactive protein,** median (IQR) mg/dl | N = 288 | 61.8 (24.2–118.6) | 60.3 (22.9–122.2) | 89.5 (33.0–135.7) | 0.279 |
| **Prothrombin time,** median (IQR) s | N = 259 | 13.2 (12.3–14.5) | 13.3 (12.4–14.6) | 12.6 (11.9–13.2) | **0.056** |

Values are presented as n (%) or mean ± SD unless otherwise stated. Statistically significant results are highlighted in bold. Abbreviations: Non-TEE: non-thromboembolic event. TEE: thromboembolic event. Reference ranges are as follows: d-dimer, 0 to 500 ng per milliliter; alanine aminotransferase (ALT), U/L <33–41; aspartate aminotransferase (AST), U/L <33–41; creatinine, mg/dL (0.7–1.2) mg/dL (0.5–0.9); hemoglobin, g/dL (12.3–15.3) (14.0–17.5); leukocytes, x10e3/µL (4.4–11.3); neutrophil count, x10e3/µL (1.5–7.5); platelet count, 150,000 to 450,000 per cubic microliter; C reactive protein, mg/L <5; prothrombin time, 9.9 to 14.2 seconds. Abbreviations: SD, standard deviation; IQR, interquartile range; ICU, intensive care unit.

group of non-survivors: hypertension (51.8% vs. 34.5%, p<0.001), diabetes mellitus (25% vs. 13.3%, p<0.001), hyperlipidemia (27.2% vs. 21.6%, p = 0.040), chronic kidney disease (8.7% vs. 2.7%, p<0.001), congestive heart failure (4.7% vs. 1.7%, p = 0.002), chronic obstructive pulmonary disease (15.6% vs. 5.2%, p<0.001), and atrial fibrillation (8.3%) (Table 2). Mortality was higher in patients initially admitted to intensive care unit than in conventional units (16.3% vs. 4.9%, p<0.001).

**Table 2. Demographic and clinical findings among COVID-19 patients with thromboembolic events, stratified by mortality.**

| | | All | Survivors | Non survivors | p-value[a] |
|---|---|---|---|---|---|
| n (%) | | 1737 | 1461 (84.1) | 276 (15.9) | - |
| **Age,** mean (SD) years | | 65.5 (16.7) | 62.6 (16.2) | 80.7 (9.7) | **<0.001** |
| **Sex female** | | 686 (39.5) | 589 (40.3) | 97 (35.1) | 0.107 |
| **Hypertension** | | 647 (37.2) | 504 (34.5) | 143 (51.8) | **<0.001** |
| **Diabetes** | | 263 (15.1) | 194 (13.3) | 69 (25.0) | **<0.001** |
| **Hyperlipidemia** | | 390 (22.5) | 315 (21.6) | 75 (27.2) | **0.040** |
| **Chronic kidney disease** | | 64 (3.7) | 40 (2.7) | 24 (8.7) | **<0.001** |
| **Congestive heart failure** | | 38 (2.2) | 25 (1.7) | 13 (4.7) | **0.002** |
| **Atrial fibrillation** | | 72 (4.1) | 49 (3.4) | 23 (8.3) | **<0.001** |
| **Previous ischemic heart disease** | | 91 (5.2) | 61 (4.2) | 30 (10.9) | **<0.001** |
| **Previous cerebrovascular disease** | | 15 (0.9) | 10 (0.7) | 5 (1.8) | 0.063 |
| **Pneumonia or respiratory symptoms** | | 1606 (92.5) | 1353 (92.6) | 253 (91.7) | 0.587 |
| **Chronic obstructive pulmonary disease** | | 119 (6.9) | 76 (5.2) | 43 (15.6) | **<0.001** |
| **Ex-smoker** | | 258 (14.9) | 213 (14.6) | 45 (16.3) | 0.460 |
| **Current smoker** | | 52 (3.0) | 45 (3.1) | 7 (2.5) | 0.627 |
| **Management** | | | | | |
| **ICU care** | | 117 (6.7) | 72 (4.9) | 45 (16.3) | **<0.001** |
| **Days of hospitalization,** median (IQR) | | 7.0 (4.0–10.0) | 7.0 (4.0–10.0) | 5.0 (3.0–9.0) | **<0.001** |
| **Thromboembolism Event** | | 64 (3.7) | 45 (3.1) | 19 (6.9) | **0.002** |
| **Venous thromboembolism** | | 49 (2.8) | 35 (2.4) | 14 (5.1) | **0.014** |
| **Acute ischemic stroke** | | 6 (0.3) | 3 (0.2) | 3 (1.1) | **0.022** |
| **Acute ischemic heart disease** | | 8 (0.5) | 5 (0.3) | 3 (1.1) | 0.094 |
| **Vital signs** | | | | | |
| **Systolic blood pressure,** mean (SD) mmHg | N = 1099 | 131.5 (21.5) | 131.9 (20.4) | 129.8 (25.9) | **<0.001** |
| **Diastolic blood pressure,** mean (SD) mmHg | N = 1104 | 76.5 (33.0) | 77.6 (35.7) | 71.3 (14.2) | 0.242 |
| **Oxygen saturation,** mean (SD) | N = 1385 | 92.3 (7.1) | 93.4 (5.3) | 86.2 (11.4) | **0.018** |
| **Laboratory determinations** | | | | | |
| **D-dimer,** median (IQR) ng/mL | N = 257 | 822.5 (450.5–1745.5) | 690.0 (403.0–1217.0) | 3321.0 (1003.5–6526.0) | **<0.001** |
| **Alanine aminotransferase,** median (IQR) U/L | N = 277 | 28.2 (17.0–49.0) | 27.4 (17.7–46.9) | 26.1 (15.5–51.1) | 0.816 |
| **Aspartate aminotransferase,** median (IQR) U/L | N = 284 | 34.0 (23.8–57.1) | 31.0 (22.8–51.9) | 46.2 (25.9–65.1) | **0.007** |
| **Creatinine,** median (IQR) mg/dL | N = 304 | 0.9 (0.7–1.1) | 0.8 (0.7–1.0) | 1.0 (0.8–1.7) | **0.001** |
| **Hemoglobin,** median (IQR) g/dL | N = 297 | 13.6 (12.2–14.8) | 13.9 (12.6–14.9) | 12.7 (11.8–13.8) | **0.001** |
| **Leukocytes count,** median (IQR) x10e3/μL | N = 308 | 6.6 (5.0–9.5) | 6.1 (4.9–8.5) | 8.4 (5.7–10.8) | **0.001** |
| **Neutrophil count,** median (IQR) x10e3/μL | N = 308 | 4.8 (3.3–7.6) | 4.4 (3.2–6.4) | 7.3 (4.1–11.1) | **<0.001** |
| **Platelets count,** median (IQR) x10$^3$ per uL | N = 302 | 218.5 (168.3–299.5) | 224.5 (171.0–311.0) | 185.0 (141.0–240.0) | **0.002** |
| **C reactive protein,** median (IQR) mg/dl | N = 288 | 61.8 (24.2–118.6) | 54.5 (21.7–107.7) | 129.7 (48.2–238.9) | **<0.001** |
| **Prothrombin time,** median (IQR) seconds | N = 259 | 13.2 (12.3–14.5) | 13.2 (12.3–14.5) | 13.8 (12.8–15.2) | **0.039** |

Values are presented as n (%) or mean ± SD unless otherwise stated. Statistically significant results are highlighted in bold. Reference ranges are as follows: d-dimer, 0 to 500 ng per milliliter; alanine aminotransferase (ALT), U/L <33–41; aspartate aminotransferase (AST), U/L <33–41; creatinine, mg/dL (0.7–1.2) mg/dL (0.5–0.9); hemoglobin, g/dL (12.3–15.3) (14.0–17.5); leukocytes, x10e3/μL (4.4–11.3); neutrophil count, x10e3/μL (1.5–7.5); platelet count, 150,000 to 450,000 per cubic microliter; C reactive protein, mg/L <5; prothrombin time, 9.9 to 14.2 seconds.

Abbreviations: SD, standard deviation; IQR, interquartile range; ICU, intensive care unit.

Monitored vital signs were also altered in non-survivor patients. Basal systolic blood pressure (mean 129.8 [SD 25.9] vs. 131.9 [SD 20.4] mmHg, p<0.001) and oxygen saturation (mean 86.2% [SD 11.4] vs. 93.4% [SD 5.3], p = 0.018) were significantly reduced in non-survivor patients.

**Table 3. Multivariate analysis and adjusted logistic regression for predictors of death.**

| Variables | Model 1 | | Model 2 | |
|---|---|---|---|---|
| | OR (CI 95%) | p-value | OR (CI 95%) | p-value |
| **Age** | 1.12 (1.10–1.14) | **<0.001** | 1.12 (1.09–1.14) | **<0.001** |
| **Diabetes** | 1.49 (1.04–2.13) | **0.029** | 2.04 (1.28–3.25) | **0.003** |
| **Chronic obstructive pulmonary disease** | 1.61 (1.03–2.53) | **0.039** | 1.81 (1.00–3.27) | **0.049** |
| **ICU care** | 9.40 (5.69–15.51) | **<0.001** | 4.86 (2.34–10.08) | **<0.001** |
| **Thromboembolism** | 2.24 (1.17–4.29) | **0.015** | 2.46 (1.11–5.45) | **0.027** |
| **Systolic blood pressure** | - | - | 0.98 (0.98–0.99) | **<0.001** |
| **Oxygen saturation** | - | - | 0.90 (0.88–0.93) | **<0.001** |

Multivariate analysis exploring the predictors of death in COVID-19 patients. Statistically significant results are highlighted in bold. Abbreviations: OR: odds ratio; CI: confidence interval.

The thrombotic profile of non-survivor patients was characterized by elevated levels of d-dimer (median 3321.0 [IQR 1003.5–6526.0] vs. 690.0 [IQR 403.0–1217.0] ng/mL, p<0.001), and prolonged prothrombin time suggesting coagulopathy (median 13.8 [IQR 12.8–15.2] vs. 13.2 [IQR 12.3–14.5] seconds, p = 0.039). The influence of inflammation and immune response was showed by elevated levels of c-reactive protein (median 129.7 [IQR 48.2–238.9] vs. 54.5 [IQR 21.7–107.7] mg/dl, p<0.001) and higher counts of leukocytes (median 8.4 [IQR 5.7–10.8] vs. 6.1 [IQR 4.9–8.5] x10e3/uL, p = 0.001) and neutrophils (median 7.3 [IQR 4.1–11.1] vs. 4.4 [3.2–6.4] x10e3/uL, p<0.001) among non-survivor patients. Finally, hemoglobin (median 12.7 [IQR 11.8–13.8] vs. 13.9 [IQR 12.6–14.9] g/dL, p = 0.001) and platelets (median 185.0 [IQR 141.0–240.0] vs. 224.5 [IQR 171.0–311.0] x10e3/uL, p = 0.002) counts were decreased in non-survivor patients (Table 2).

The multivariate regression models confirmed the association between in-hospital death and age (odds ratio [OR] 1.12 [95% CI 1.10–1.14], p<0.001), diabetes (OR 1.49 [95% CI 1.04–2.13], p = 0.029), chronic obstructive pulmonary disease (OR 1.61 [95% CI 1.03–2.53], p = 0.039), ICU care (OR 9.39 [95% CI 5.69–15.51], p<0.001), and thromboembolism presence (OR 2.24 [95% CI 1.17–4.29], p = 0.015). In addition, basal systolic blood pressure (OR 0.98 [95% CI 0.98–0.99], p<0.001) and basal oxygen saturation (OR 0.90 [95% CI 0.88–0.93], p<0.001) were inversely related to the risk of in-hospital mortality (Table 3).

## Discussion

Our results showed a deleterious effect of TEE, age, diabetes mellitus, and COPD in in-hospital mortality among COVID-19 patients (Table 3). Previous reports identified both the relationship of COVID-19 infection and TEE [5, 6, 21–23] and the increased risk of death among TEE patients [24]. However, most of them have been limited in size and focused on patients with severe disease hospitalized in intensive care units. These aspects could explain the significantly lower observed TEE risk in our investigation than in others. We have confirmed the association of D-dimer levels and TEE events [24]. In initial reports of COVID-19 infection, D-dimers levels and prolonged prothrombin time have defined a coagulation disorder associated with COVID-19 infection severity [1]. In line with this hypothesis, recent autopsy studies identified extensive extracellular fibrin deposition and the presence of fibrin thrombi within small vessels and capillaries of the lungs [25–27]. As in our study, VTE is the most described thrombotic complication [6, 28]. COVID-19 infection was also associated with hyperviscocity, which could explain anticoagulation failures [29]. Moreover, in cases of sepsis the overproduction of proinflammatory cytokines can cause microvasculature and endothelial dysfunction,

which could trigger hemostatic cascade [30]. In addition, the prothrombotic effect of COVID-19 infection might be related with the risk of developing acute coronary syndrome, that increases on acute infections like influenza epidemics [31–33] due to the increment of myocardial demands triggered by infections [34]. Also, there is an evidence of increasing in-hospital mortality among patients with myocardial injury associated with elevated troponin T levels [35, 36]. As well, a direct viral infection of vascular endothelium and myocardium is possible [37, 38]. Stroke mechanisms include hypercoagulability and cardioembolic stroke of cited virus-related with cardiac injury [10]. The absence of association between general risk factors for TEE in our cohort was similar with previous studies [6, 39].

As in other respiratory viral infections [31, 40], we observed that old patients [7, 41, 42] and COPD [43] were associated with a higher risk of poor outcomes or death. Recent data indicate that diabetes is an important risk factor for unfavorable outcome in COVID-19 patients. However, most of the studies described a small number of cases [7, 19, 42]. Our investigation included enough patients to demonstrated independent predictive value of this risk factor on mortality. This relationship could be partially explained because diabetic patients are predisposed to a hyper-inflammatory and pro-coagulant state [44]. COVID-19 infection could cause pleiotropic alterations of glucose metabolism due to direct effect on angiotensin-converting enzyme 2 (ACE2) receptors of pancreatic beta cells [45].

Our study has some important limitations as well. First, the retrospective nature of the study might constraint the dataset. Second, we analyzed data supplied by HM Hospitales and it would have been beneficial to perform a revision of the electronic medical records of the included patients, concretely the TEE group to refine some of the variables and get a deeper description. Third, information of previous treatments like anticoagulants or antiplatelet agents was not available. Fourth, we were not able to evaluate a relevant cofounder such as the motivation of hospitalization (clinical or surgical). These limitations avoid us making a recommendation of thrombosis prophylaxis although, due to the large number of patients included we considered that our sample is representative of a larger population. A recent meta-analysis concluded that there is currently insufficient evidence to determine the risks and benefits of prophylactic anticoagulants for people hospitalized with COVID-19 [46].

In conclusion, the analysis of this multicenter retrospective observational dataset of COVID-19 patients confirmed some of the previous observations related with TEE in COVID-19 patients. The prevalence of TEE was not negligible. In addition, patients who develop TEE such as patients with other comorbidities as diabetes, COPD, and older patients had worse prognosis than patient without TEE or theses comorbidities.

## Supporting information

**S1 Checklist. STROBE statement—checklist of items that should be included in reports of *cohort studies.***
(DOC)

## Acknowledgments

We thank HM Hospitales for making their data publicly available as part of the COVID Data Save Lives project.

## Author Contributions

**Conceptualization:** Francisco Purroy, Gloria Arqué.

**Data curation:** Francisco Purroy.

**Formal analysis:** Francisco Purroy, Gloria Arqué.

**Investigation:** Francisco Purroy, Gloria Arqué.

**Methodology:** Francisco Purroy, Gloria Arqué.

**Project administration:** Francisco Purroy.

**Software:** Francisco Purroy.

**Supervision:** Francisco Purroy, Gloria Arqué.

**Validation:** Francisco Purroy.

**Visualization:** Francisco Purroy.

**Writing – original draft:** Francisco Purroy, Gloria Arqué.

**Writing – review & editing:** Francisco Purroy, Gloria Arqué.

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
