## [Decision Letter · Decision Letter 0]

26 Mar 2021

PONE-D-21-03654

Influence of thromboembolic events in the prognosis of COVID-19 hospitalized patients

PLOS ONE

Dear Dr. Purroy,

Thank you for submitting your manuscript to PLOS ONE. After careful consideration, we feel that it has merit but does not fully meet PLOS ONE’s publication criteria as it currently stands. Therefore, we invite you to submit a revised version of the manuscript that addresses the points raised during the review process.

We look forward to receiving your revised manuscript.

Kind regards,

Aleksandar R. Zivkovic

Academic Editor

PLOS ONE

Journal Requirements:

2. Thank you for stating in the text of your manuscript "The study was approved by the Ethics committee of the HM Hospitals and adhered to the policy for protection of human subjects according to Declaration of Helsinki. As the information was obtained from a de-identified public database written informed consent from all study participants was not needed."

Please also add this information to your ethics statement in the online submission form.

3. Thank you for providing the date(s) when patient medical information was initially recorded.

Please also include the date(s) on which your research team accessed the databases/records to obtain the retrospective data used in your study.

4. In your Methods section, please provide additional information about the participant selection method and the demographic details of your participants. Please ensure you have provided sufficient details to replicate the analyses such as:

a) a list of the sixteen hospitals that make up the HM Hospitales

b) description of any inclusion/exclusion criteria that were applied to participant selection (e.g., ICD10 codes, characteristics, age, etc.)

c) a statement as to whether your sample can be considered representative of a larger population

5.Thank you for stating in your financial disclosure: 

'The funders had no role in study design, data collection and analysis, decision to publish, or preparation of the manuscript.'

Reviewers' comments:

Reviewer's Responses to Questions

Review Comments to the Author

Reviewer #1: I appreciated the asking for a peer review report, and I give kind regards for the whole work. All my comments and suggestions are intending to improve the quality of the work.

Specific comments

page 1, title. The authors should follow the journal recommendations about manuscript format (https://journals.plos.org/plosone/s/submission-guidelines) and follow the STROBE checklist when formatting your text. For instance, from what I could see, it is a non-randomized observational study, but it was not clearly stated in the title and the abstract.

Is there a registration number to be reported? If yes, please provide the number here.

Additionally, submitting the fulfilled STROBE checklist accompanying your manuscript is highly recommended.

pages1-2 financial disclosure. The authors state that 'The funders had no role in study design, data collection and analysis, decision to publish, or preparation of the manuscript.' but there is no detail about the received found. Please provide the details (Initials of the authors who received each award / Grant numbers awarded to each author / The full name of each funder / URL of each funder website) as required by the journal.

Page 3, ethics statement. the authors state that 'HM Hospital ethics comittee' but there is no detail about ethics statement. Please, provide details (Give the name of the institutional review board or ethics committee that approved the study / • Include the approval number and/or a statement indicating approval of this research / • Indicate the form of consent obtained (written/oral) or the reason that consent was not obtained (e.g. the data were analyzed anonymously)) as required by the journal.

Page 8, keywords. The authors have used some mesh terms (https://www.ncbi.nlm.nih.gov/mesh/) as 'COVID-19'. However, I suggest using other Mesh terms in the keywords section to improve the article indexing and citation, such as 'venous thrombosis' instead of deep vein thrombosis and 'thrombosis' instead of thromboembolic disease.

page 8. abbreviations. Please, could you explain the exact differences between composite thromboembolic outcome and thromboembolism event terms? I believe that they are very similar and could be merged in a unique term.

pages9, abstract. the abstract text should reflect the same information as the full text; for instance, the methods used should be described here and not only the primary outcome of interest (incidence of TEE). Besides, the conclusion should primarily answer your objectives and not make inferences or recommendations. Please, amend your conclusions.

page 12, methods. The authors state that the main objective was to determine the composite outcome incidence, but there is no description of follow-up time. To establish an incidence, you must follow the cohort for some time and preferable report the results as a risk ratio. However, you did not report the time of follow-up. In truth, it seems that it was a cross-sectional study that used a de-identified database. If you performed a cross-sectional study, you could not report any incidence data, only prevalence. Please, clarify your methods.

page 14, line 155. Please clarify if the numbers are correct: 48 participants represent 75%?

Page 14, results. please, could you explain the difference between composite thromboembolic outcome and thromboembolism event terms? Please, see my comments on page 8 above. The authors describe a median time (7 days) until discharge, but there was 15.9% mortality. What was the time until death?

page 18, discussion. The authors did not evaluate relevant confounders such as the motivation of hospitalization (clinical or surgical), the previous use of anticoagulants or antiplatelet agents, the need and the dose of used anticoagulants the participants during hospitalization. Although all hospitalized patient should undergo a checklist verification about the necessity of thrombosis prophylaxis, you can not make this general recommendation for all people with COVID-19, based on your evidence. Besides, there is a lack of evidence for anticoagulant therapy in people hospitalized with COVID-19 (please see https://pubmed.ncbi.nlm.nih.gov/33502773/). Please amend your last sentence.

page 18, conclusion. Clinical decision making should be done under a synthesis of the best available evidence and an evaluation of this evidence's certainty. An observational study that did not evaluate previous and current anticoagulation without a specific follow-up period does not allow you to make any recommendation regarding 'thrombosis prophylaxis in related patients. The conclusion should primarily answer your objectives and not make inferences or recommendations. Please revise your conclusions to strict answer your objectives, as the good scientific practice recommends.

Reviewer #2: Well written article. The incidence of thrombotic events in critically ill COVID-19 infected patients is markedly high as compared to critically ill non-COVID-19 infected patients is markedly high. While the venous thromboses appear in proliferation of publications, the data about the incidence of arterial and capillary thromboses are still scant. While the authors have addressed most of the points accurately in their manuscript, I have a few concerns

Minor Points:

# This is a thoughtful and intriguing study, however, many studies published during this pandemic talk about thromboses in COVID infected patients. How is this study different from the rest of them?? or what is it, in this study that separates and distinguishes it from the already published literature has not been made clear by the authors.

# 'Results' section of this manuscripts gives statistics of the study in very fine details. However, it becomes difficult to maintain the flow while reading - if simplified further, maybe by adding a couple of paragraphs, will be a good addition to this paper.

# Please mention the inclusion and exclusion criteria clearly in the material and methods section.

# A flowchart depicting the study population, excluded and included patients will be a good addition to this paper

# Line no. 49 - 'Emerging' will be a better word than 'on going' in this place

In conclusion, this is an interesting study. If worked on the above points, it will only add to its quality

Reviewer #3: Formal comment:

Formal comment to Franciscoet al.: Influence of thromboembolic events in the prognosis of COVID-19 hospitalized patients.

Chun-yi Wang,MB, Wen Wen,MB, Jie Ni,MB, Jing-jie Jiang,MB, Ming-Wei Wang,MD,PhD*,Guofan Chen,MD,PhD*

Affiliated Hospital of Hangzhou Normal University , Hangzhou,310015,China

Chun-yi Wang and Wen Wen contributed equally to this work

*Corresponding Author: Ming-Wei Wang; E-mail: wmw990556@163.com;Guofan Chen,E-mail: 495086736@qq.com

Author Contributions

Writing–original draft: Chun-yi Wang,Wen Wen,Jie Ni and Jing-jie Jiang.

Writing–review & editing: Ming-wei Wang and Guofan Chen.

We read with great interest an article published in PLoS ONE titled “Influence of thromboembolic events in the prognosis of COVID-19 sickened patients.” [1] On the basis of a study conducted among patients with COVID-19 at 16 hospitals in Spain, the authors concluded that COVID-19 patients experiencing thrombotic events had significantly poorer prognoses than those without thrombotic events. They also confirmed higher mortality rates in patients who have developed complications COVID-19 than those without thrombotic events.

Although the author strengthened the correlation between thrombotic events and prognosis in patients through this study and proved the need for clinical thrombotic prevention and antithrombotic treatment, several points need to be considered in the interpretation of the presented results. Composite thromboembolic outcome (CTE) was defined as arterial thromboembolic complications and venous thromboembolic complications by the author.

Venous thromboembolism (VTE), clinically manifested as deep vein thrombosis or pulmonary embolism, is the third most common acute cardiovascular syndrome after myocardial infarction and stroke worldwide [2]. Endothelial injury, hypercoagulability, and blood stasis (Virchow factors) are the three risk factors for thrombosis. Infection is a common trigger for VTE. Acute infection was assigned 1 point under the Padua Prediction Score [3].

In our meta-analysis, we concluded that the prognosis of patients with VTE was significantly worse than that of patients without thrombus; moreover, the disease was more severe in patients with VTE than in those without thrombus [4], which was consistent with the results of the author's study.

A meta-analysis by Tan BK et al. mentioned that patients with severe COVID-19 had a higher risk of VTE on admission. A certain risk of arterial embolization was present, but relevant data remained inadequate [5].

However, our study mainly focused on VTE and did not include the outcome of arterial thrombotic events in patients with COVID-19. McBane RD 2nd argued that most reports focused on VTE, and few studies provided the incidence of arterial thrombotic events, which varied from 2% to 5% [6]. Fournier M et al. also analyzed arterial thrombosis, but the prevalence remained relatively low [7].

The prevalence of arterial thrombotic events (acute ischemic stroke, systemic arterial embolism, and myocardial infarction) considered by the authors was low. CTE was mentioned in 64 cases, but 76.6% of them were VTE [1].

Arterial thrombus formation generally occurs because of certain underlying diseases or certain inducements or risk factors. For instance, acute thrombotic events may occur with coronary stents after percutaneous transluminal coronary intervention for acute myocardial infarction. However, acute arterial embolism occurs because as a “foreign body,” the stent induces platelet aggregation. The origin of arterial thrombosis is thus difficult to determine.

Therefore, whether the authors ignore arterial thrombotic events and only consider studying venous thrombosis has to be determined. On the basis of the present study, the correlation between arterial thrombosis and venous thrombosis in patients with COVID-19 may be separately studied. Regardless of the aforementioned point, we agree with the author's view that patients with COVID-19 experiencing thrombotic events have poor prognoses and increased mortality. Therefore, during clinical treatment, antithrombotic therapy should be administered in patients with COVID-19.

6. PLOS authors have the option to publish the peer review history of their article (what does this mean?). If published, this will include your full peer review and any attached files.

Reviewer #1: **Yes: **Ronald Luiz Gomes Flumignan, MD, PhD

Reviewer #2: No

Reviewer #3: **Yes: **Mingwei Wang

---

## [Author Response · Author response to Decision Letter 0]

9 May 2021

Dear Aleksandar R. Zivkovic, 

Thank you for giving us the opportunity to resubmit a revised draft of our manuscript titled “Influence of thromboembolic events in the prognosis of COVID-19 hospitalized patients” to your prestigious journal Plos One. We appreciate the time and effort that you and the reviewers have dedicated to providing your valuable feedback on our manuscript. We are grateful to the reviewers for their insightful comments on the paper. We have been able to incorporate changes to reflect most of the suggestions from the reviewers. We have highlighted the changes within the manuscript using track changes.

Here is a point-by-point response to the editors and reviewers’ comments.

Responses to Academic Editor

Comment 1. Please ensure that your manuscript meets PLOS ONE's style requirements, including those for file naming. The PLOS ONE style templates can be found at

Response: We have reviewed PLOS ONE’s style requirements to improve the original submission.

Comment 2. Thank you for stating in the text of your manuscript "The study was approved by the Ethics committee of the HM Hospitals and adhered to the policy for protection of human subjects according to Declaration of Helsinki. As the information was obtained from a de-identified public database written informed consent from all study participants was not needed."

Please also add this information to your ethics statement in the online submission form.

Response: We have considered editor’s comment

Comment 3. Thank you for providing the date(s) when patient medical information was initially recorded.

Please also include the date(s) on which your research team accessed the databases/records to obtain the retrospective data used in your study.

Response: We have addressed the editor’s comment in the new version of the manuscript.

Comment 4. In your Methods section, please provide additional information about the participant selection method and the demographic details of your participants. Please ensure you have provided sufficient details to replicate the analyses such as:

a) a list of the sixteen hospitals that make up the HM Hospitales

b) description of any inclusion/exclusion criteria that were applied to participant selection (e.g., ICD10 codes, characteristics, age, etc.)

c) a statement as to whether your sample can be considered representative of a larger population

Response: We have addressed the editor’s comments in the new version of the manuscript. We have listed all HM hospitals. We have clarified that patients under 18 years old were excluded. In addition, patients without ICD10 codification were also excluded (n=131). We add an statement in the discussion section highlighting that our sample could be considered representative of a larger population.

Comment 5. Thank you for stating in your financial disclosure: 

'The funders had no role in study design, data collection and analysis, decision to publish, or preparation of the manuscript.'

- Please clarify the sources of funding (financial or material support) for your study. List the grants or organizations that supported your study, including funding received from your institution.

- State what role the funders took in the study. If the funders had no role in your study, please state: “The funders had no role in study design, data collection and analysis, decision to publish, or preparation of the manuscript.”

- If any authors received a salary from any of your funders, please state which authors and which funders.

- If you did not receive any funding for this study, please state: “The authors received no specific funding for this work.”

Response: We have add information about the founding in the new version of the manuscript. We confirm that we received no specific funding for this work.

Comment 6. Please review your reference list to ensure that it is complete and correct. If you have cited papers that have been retracted, please include the rationale for doing so in the manuscript text or remove these references and replace them with relevant current references. Any changes to the reference list should be mentioned in the rebuttal letter that accompanies your revised manuscript. If you need to cite a retracted article, indicate the article’s retracted status in the References list and also include a citation and full reference for the retraction notice.

Response: We have reviewed all cites. We confirm that no retracted article has been added in the bibliography of the new version of the article. 

 

Responses to Reviewer #1

I appreciated the asking for a peer review report, and I give kind regards for the whole work. All my comments and

suggestions are intending to improve the quality of the work.

Response: We appreciate the time and effort that you have dedicated to providing your valuable feedback on our manuscript.

Comment 1

page 1, title. The authors should follow the journal recommendations about manuscript format (https://journals.plos.org/plosone/s/submission-guidelines) and follow the STROBE checklist when formatting your text. For instance, from what I could see, it is a non-randomized observational study, but it was not clearly stated in the title and the abstract.

Is there a registration number to be reported? If yes, please provide the number here.

Additionally, submitting the fulfilled STROBE checklist accompanying your manuscript is highly recommended.

Response: We have taken into account reviewer 1 comment in the new version of the manuscript. We clarify that is a study based on a cross sectional study in the title and in the abstract. We have added a fulfilled STROBE checklist. 

Comment 2

pages1-2 financial disclosure. The authors state that 'The funders had no role in study design, data collection and analysis, decision to publish, or preparation of the manuscript.' but there is no detail about the received found. Please provide the details (Initials of the authors who received each award / Grant numbers awarded to each author / The full name of each funder / URL of each funder website) as required by the journal.

Response: We have clarified this point in the new version of the manuscript. We confirm that we received no specific funding for this work.

Comment 3

Page 3, ethics statement. the authors state that 'HM Hospital ethics comittee' but there is no detail about ethics statement. Please, provide details (Give the name of the institutional review board or ethics committee that approved the study / • Include the approval number and/or a statement indicating approval of this research / • Indicate the form of consent obtained (written/oral) or the reason that consent was not obtained (e.g. the data were analyzed anonymously)) as required by the journal.

Response: We have taken into account reviewer 1 comment in the new version of the manuscript. The study was approved by the Ethics committee of the HM Hospitals (approval number 20.03.1573-GHM).and adhered to the policy for protection of human subjects according to Declaration of Helsinki. As the information was obtained from a de-identified public database written informed consent for our specific study from all study participants was not needed. The data were analyzed anonymously. 

Comment 4

Page 8, keywords. The authors have used some mesh terms (https://www.ncbi.nlm.nih.gov/mesh/) as 'COVID-19'. However, I suggest using other Mesh terms in the keywords section to improve the article indexing and citation, such as 'venous thrombosis' instead of deep vein thrombosis and 'thrombosis' instead of thromboembolic disease.

Response: We have taken into account reviewer 1 comment in the new version of the manuscript.

Comment 5

page 8. abbreviations. Please, could you explain the exact differences between composite thromboembolic outcome and thromboembolism event terms? I believe that they are very similar and could be merged in a unique term.

Response: We have taken into account reviewer 1 comment in the new version of the manuscript. We use only one term in the new version of the manuscript. 

Comment 6

pages9, abstract. the abstract text should reflect the same information as the full text; for instance, the methods used should be described here and not only the primary outcome of interest (incidence of TEE). Besides, the conclusion should primarily answer your objectives and not make inferences or recommendations. Please, amend your conclusions.

Response: We have taken into account reviewer 1 comment in the new version of the manuscript.

Comment 7

page 12, methods. The authors state that the main objective was to determine the composite outcome incidence, but there is no description of follow-up time. To establish an incidence, you must follow the cohort for some time and preferable report the results as a risk ratio. However, you did not report the time of follow-up. In truth, it seems that it was a cross-sectional study that used a de-identified database. If you performed a cross-sectional study, you could not report any incidence data, only prevalence. Please, clarify your methods.

Response: We have taken into account reviewer 1 comment in the new version of the manuscript. We confirm that we did a cross-sectional study that used a de-identified database. We recognize that we could no offer information about incidence. 

Comment 8

page 14, line 155. Please clarify if the numbers are correct: 48 participants represent 75%?

Response: We have taken into account reviewer 1 comment in the new version of the manuscript. We have clarified that these 48 participants represented 75.0% of the 64 TEE patients

Comment 9

Page 14, results. please, could you explain the difference between composite thromboembolic outcome and thromboembolism event terms? Please, see my comments on page 8 above. The authors describe a median time (7 days) until discharge, but there was 15.9% mortality. What was the time until death?

Response: We have taken into account reviewer 1 comment in the new version of the manuscript. We agree with reviewer 1 that is better to use one term. Median time until death was 5.0 days (IQR 3.0-9.0) days. We have added this information in the new version of the manuscript. 

Comment 10

page 18, discussion. The authors did not evaluate relevant confounders such as the motivation of hospitalization (clinical or surgical), the previous use of anticoagulants or antiplatelet agents, the need and the dose of used anticoagulants the participants during hospitalization. Although all hospitalized patient should undergo a checklist verification about the necessity of thrombosis prophylaxis, you can not make this general recommendation for all people with COVID-19, based on your evidence. Besides, there is a lack of evidence for anticoagulant therapy in people hospitalized with COVID-19 (please see https://pubmed.ncbi.nlm.nih.gov//). Please amend your last sentence.

Response: We have considered reviewer 1 comment in the new version of the manuscript. We have added the limitations highlighted in the discussion. We have avoided a general recommendation for all people with COVID-19 infection. We have amended our previous last sentence in the new version of the manuscript.

Comment 11

page 18, conclusion. Clinical decision making should be done under a synthesis of the best available evidence and an evaluation of this evidence's certainty. An observational study that did not evaluate previous and current anticoagulation without a specific follow-up period does not allow you to make any recommendation regarding 'thrombosis prophylaxis in related patients. The conclusion should primarily answer your objectives and not make inferences or recommendations. Please revise your conclusions to strict answer your objectives, as the good scientific practice recommends.

Response: We have taken into account reviewer 1 comment in the new version of the manuscript. 

“In conclusion, the analysis of this multicenter retrospective observational dataset of COVID-19 patients confirmed some of the previous observations related with TEE in COVID-19 patients. The prevalence of TEE was not negligible. In addition, patients who develop TEE such as patients with other comorbidities as diabetes, COPD, and older patients had worse prognosis than patient without TEE or theses comorbidities.”

Responses to Reviewer #2

Reviewer #2: Well written article. The incidence of thrombotic events in critically ill COVID-19 infected patients is markedly high as compared to critically ill non-COVID-19 infected patients is markedly high. While the venous thromboses appear in proliferation of publications, the data about the incidence of arterial and capillary thromboses are still scant. While the authors have addressed most of the points accurately in their manuscript, I have a few concerns

Minor Points:

Comment 1

This is a thoughtful and intriguing study, however, many studies published during this pandemic talk about thromboses in COVID infected patients. How is this study different from the rest of them?? or what is it, in this study that separates and distinguishes it from the already published literature has not been made clear by the authors.

Response: We recognized that there are many studies published during this pandemic about thromboses. We think that one of our strengths is the size of the study. This issue allow us to consider that our sample is representative of a larger population. 

Comment 2

'Results' section of this manuscripts gives statistics of the study in very fine details. However, it becomes difficult to maintain the flow while reading - if simplified further, maybe by adding a couple of paragraphs, will be a good addition to this paper.

Response: According to reviewer 2’ comment we have re-written the results section. 

Comment 3

Please mention the inclusion and exclusion criteria clearly in the material and methods section.

Response: According to reviewer 2’ comment we have described the inclusion and exclusion criteria 

Comment 4

A flowchart depicting the study population, excluded and included patients will be a good addition to this paper

Response: According to reviewer 2’ comment we have added a flowchart depicting the study population.

Comment 5

Line no. 49 - 'Emerging' will be a better word than 'on going' in this place

Response: We have considered reviewer 2 comment in the new version of the manuscript

Responses to Reviewer #3

Comment 

Venous thromboembolism (VTE), clinically manifested as deep vein thrombosis or pulmonary embolism, is the third most common acute cardiovascular syndrome after myocardial infarction and stroke worldwide [2]. Endothelial injury, hypercoagulability, and blood stasis (Virchow factors) are the three risk factors for thrombosis. Infection is a common trigger for VTE. Acute infection was assigned 1 point under the Padua Prediction Score [3]. In our meta-analysis, we concluded that the prognosis of patients with VTE was significantly worse than that of patients without thrombus; moreover, the disease was more severe in patients with VTE than in those without thrombus [4], which was consistent with the results of the author's study.

A meta-analysis by Tan BK et al. mentioned that patients with severe COVID-19 had a higher risk of VTE on admission. A certain risk of arterial embolization was present, but relevant data remained inadequate [5].

However, our study mainly focused on VTE and did not include the outcome of arterial thrombotic events in patients with COVID-19. McBane RD 2nd argued that most reports focused on VTE, and few studies provided the incidence of arterial thrombotic events, which varied from 2% to 5% [6]. Fournier M et al. also analyzed arterial thrombosis, but the prevalence remained relatively low [7]. The prevalence of arterial thrombotic events (acute ischemic stroke, systemic arterial embolism, and myocardial infarction) considered by the authors was low. CTE was mentioned in 64 cases, but 76.6% of them were VTE [1]. 

Arterial thrombus formation generally occurs because of certain underlying diseases or certain inducements or risk factors. For instance, acute thrombotic events may occur with coronary stents after percutaneous transluminal coronary intervention for acute myocardial infarction. However, acute arterial embolism occurs because as a “foreign body,” the stent induces platelet aggregation. The origin of arterial thrombosis is thus difficult to determine.

Therefore, whether the authors ignore arterial thrombotic events and only consider studying venous thrombosis has to be determined. On the basis of the present study, the correlation between arterial thrombosis and venous thrombosis in patients with COVID-19 may be separately studied. Regardless of the aforementioned point, we agree with the author's view that patients with COVID-19 experiencing thrombotic events have poor prognoses and increased mortality. Therefore, during clinical treatment, antithrombotic therapy should be administered in patients with COVID-19.

Response: We have considered reviewer’s comments in the new version of the manuscript. 

We have defined better the primary outcome of the study in the abstract and in the text. 

TEE was defined by the occurrence of venous thromboembolism (VTE), acute ischemic stroke (AIS), systemic arterial embolism or myocardial infarction (MI).

We agree that it would be interesting to have information about treatments. We add this aspect in the limitation section. 

Our study has some important limitations as well. First, the retrospective nature of the study might constraint the dataset. Second, we analyzed data supplied by HM Hospitales and it would have been beneficial to perform a revision of the electronic medical records of the included patients, concretely the TEE group to refine some of the variables and get a deeper description. Third, information of previous treatments like anticoagulants or antiplatelet agents was not available. Fourth, we were not able to evaluate a relevant cofounder such as the motivation of hospitalization (clinical or surgical). These limitations avoid us making a recommendation of thrombosis prophylaxis although, due to the large number of patients included we considered that our sample is representative of a larger population. A recent meta-analysis concluded that there is currently insufficient evidence to determine the risks and benefits of prophylactic anticoagulants for people hospitalized with COVID-19

---

## [Editor Report · Decision Letter 1]

14 May 2021

Influence of thromboembolic events in the prognosis of COVID-19 hospitalized patients. Results from a cross sectional study

PONE-D-21-03654R1

Dear Dr. Purroy,

We’re pleased to inform you that your manuscript has been judged scientifically suitable for publication and will be formally accepted for publication once it meets all outstanding technical requirements.

Kind regards,

Aleksandar R. Zivkovic

Academic Editor

PLOS ONE

---

## [Editor Report · Acceptance letter]

25 May 2021

PONE-D-21-03654R1 

Influence of thromboembolic events in the prognosis of COVID-19 hospitalized patients. Results from a cross sectional study. 

Dear Dr. Purroy:

I'm pleased to inform you that your manuscript has been deemed suitable for publication in PLOS ONE. Congratulations! Your manuscript is now with our production department. 

Kind regards, 

on behalf of

Dr. Aleksandar R. Zivkovic 

Academic Editor

PLOS ONE